# SlerpFlow: Spherical Trajectory Correction for Rectified Flow Inversion

**Wenbin Duan** [1] **Yan Shu** [2] **Zhuoyuan Fu** [1] **Fangmin Zhao** [3] **Yan Li** [1] **Yaru Zhao** [1] **Binyang Li** [1]

## Abstract

Rectified-flow-based diffusion transformers, particularly FLUX, have demonstrated outstanding performance in high-quality image generation. However, achieving fast and accurate inversion—transforming images back to latent noise for faithful reconstruction and editing—remains a challenging bottleneck due to the discretization errors of linear solvers. This paper introduces **SlerpFlow**, a straightforward yet highly effective zero-shot approach that unlocks the full potential of FLUX for high-fidelity inversion and editing. Unlike existing approaches (e.g., RF-Solver) that rely on complex numerical approximations such as high-order Taylor expansions to correct trajectory errors, we present a geometric view based on the Manifold Hypothesis: the empirically observed trajectory curvature is not a numerical artifact, but rather serves as a necessary "centripetal force" that constrains the flow to remain on the data manifold. Guided by this insight, SlerpFlow integrates Spherical Linear Interpolation (Slerp) to rectify flow velocity directions on the hypersphere, strictly adhering to the intrinsic curvature of the latent space. Crucially, by caching the corrected velocity for subsequent steps, SlerpFlow achieves high-precision inversion while maintaining the computational efficiency of a first-order Euler solver. Extensive experiments on FLUX-based reconstruction and editing tasks demonstrate that SlerpFlow improves reconstruction fidelity and achieves stronger semantic alignment in editing without requiring additional training. Code is available at this URL.

[1]University of International Relations, Beijing, China [2]University of Trento, Trento, Italy [3]Institute of Information Engineering, Chinese Academy of Sciences, Beijing, China. Correspondence to: Yan Li <liyan@uir.edu.cn>.

*Proceedings of the 43rd International Conference on Machine Learning*, Seoul, South Korea. PMLR 306, 2026. Copyright 2026 by the author(s).

## 1. Introduction

Rectified Flow (RF) and Flow Matching (FM) establish a deterministic paradigm for generative modeling by learning a continuous-time transport from a simple noise prior to the data distribution (Chen et al., 2018; Lipman et al., 2023; Liu et al., 2023; Song et al., 2021b). A key implication is *theoretical reversibility*: in principle, integrating the learned dynamics backward should recover a latent representation for a given real image, while integrating forward from that latent should reconstruct the image (Starodubcev et al., 2024; Lee et al., 2024; Song et al., 2021b; Grathwohl et al., 2019). This property underpins a broad class of inversion-based applications that require trajectory-level manipulation, including semantic editing (Mokady et al., 2023; Hertz et al., 2023; Wallace et al., 2023).

However, accurate and robust inversion remains challenging, and inversion quality is often the key bottleneck that determines the fidelity, consistency, and controllability of downstream applications. A canonical baseline for diffusion-model inversion is DDIM inversion (Song et al., 2021a). It performs inversion via a deterministic recurrence, where the state is updated at each timestep using the network predictor, most commonly the noise prediction $\epsilon_\theta(x, t)$. Analogously, for flow-matching models, inversion amounts to integrating the learned velocity field $v_\theta(x_t, t)$ backward in time (Lipman et al., 2023; Lee et al., 2024; Chen et al., 2018; Liu et al., 2023). Consequently, inversion quality is tightly coupled to both the accuracy of the predicted velocity and the numerical stability of its discretized integration(Zhou & Liu, 2025): Approximation errors or solver errors can accumulate across timesteps, ultimately degrading reconstruction fidelity and limiting the consistency and controllability of downstream tasks.

To alleviate this error accumulation, (Wang et al., 2025) propose RF-Solver, a training-free sampler that reduces ODE-solving errors by deriving an exact formulation of the rectified-flow ODE and applying high-order Taylor expansions to better approximate its nonlinear components, thereby improving inversion precision at each timestep. However, under the manifold hypothesis, meaningful latent representations concentrate in a tubular neighborhood of a low-dimensional manifold $\mathcal{M} \subset \mathbb{R}^D$ (Fefferman et al., 2016; Bengio et al., 2013). Flow Matching and Rectified

Flow learn an ambient-space velocity field $v_\theta(z, t)$ whose ODE transports probability mass from a simple prior to the data distribution (Lipman et al., 2023; Liu et al., 2023).In this view, the learned trajectories are expected to remain close to $\mathcal{M}$ throughout the transport (Fefferman et al., 2016; Niyogi et al., 2011; Balakrishnan et al., 2012) since moving in the normal direction typically leaves the high-density region around the data manifold (Stanczuk et al., 2024) Motivated by the manifold hypothesis, we argue that the empirically observed trajectory curvature is not merely a discretization error. Instead, it acts as a necessary "centripetal force" that counteracts normal drift and keeps the transport path within the high-density tubular neighborhood of the data manifold $\mathcal{M}$.

Building on this interpretation, we design **SlerpFlow** to *respect* the induced geometric force rather than eliminate it: instead of correcting trajectories with increasingly high-order *Euclidean* expansions, we enforce a manifold-consistent update that advances along the local tangent direction while explicitly suppressing normal drift, thereby keeping iterates inside the high-density tubular neighborhood of the data manifold $\mathcal{M}$. Concretely, we decouple the latent dynamics into radial and angular components and redefine each integration step through the lens of intrinsic geometry; in high-dimensional generative flows where probability mass concentrates near thin shells(Vershynin, 2018), we argue that minimizing *angular discretization error* is often more critical for perceptual fidelity than strictly adhering to Euclidean straight-line updates. To this end, **SlerpFlow** constructs a discrete chordal approximation: given the current state $Z_t$, it first predicts the next-step radial shell $\rho_{t+h}$ together with the target direction $\mathbf{d}_{t+h}$ (via spherical linear interpolation, SLERP), then synthesizes an effective secant (chord) vector connecting $Z_t$ to this predicted target point, resulting in the update $Z_{t+h} = Z_t + h\, v_{\text{slerp}}$.By construction, this chordal step lands *exactly* on the predicted radial shell, yielding structural exactness in the radial component and reduced geodesic deviation in the angular component, which together eliminate spurious "centrifugal" drift and enable high-fidelity inversion without increasing the number of function evaluations (NFE).

Our contributions are summarized as follows:

- We theoretically decompose the discretization error of explicit solvers, isolating the spurious centrifugal drift that plagues high-dimensional generative flows.

- We propose SlerpFlow, which implements a Decoupled Chordal Update. It guarantees that the trajectory strictly adheres to the prescribed radial dynamics.

- SlerpFlow achieves state-of-the-art reconstruction quality without requiring additional function evaluations

(NFE), validating the efficacy of the quasi-spherical geometric prior.

## 2. Related Work

### 2.1. Inversion

Inversion bridges the real data distribution and the latent noise space, serving as the cornerstone for faithful reconstruction and editing. DDIM inversion (Song et al., 2021a) approximates the reverse SDE with a deterministic ODE. However, the non-linear noise schedule introduces significant discretization errors at limited steps. To mitigate these discretization errors and bridge the discrepancy between the forward and reverse trajectories, substantial research efforts have been dedicated (Mokady et al., 2023; Wallace et al., 2023; Dong et al., 2023; Miyake et al., 2025; Bao et al., 2025; Samuel et al., 2025; Rout et al., 2024) With the rise of flow-based models, inversion has shifted from solving stochastic differential equations (SDEs) to ordinary differential equations,Recent approaches have been:RF-Inversion (Rout et al., 2025) applies dynamic optimal control to minimize global trajectory deviation. RF-Solver (Wang et al., 2025) utilizes Taylor expansions to correct the drift, while FireFlow (Deng et al., 2025) leverages gradient information from previous steps to achieve second-order accuracy with only one function evaluation (NFE) per step.

### 2.2. Editing

A widely adopted paradigm for real image editing in diffusion models is invert-then-edit: one first maps a real image back to a latent/noise state that can be faithfully reconstructed, and then performs conditional generation while injecting source information to preserve structure. Early training-free methods such as Prompt-to-Prompt control word-level semantics through cross-attention manipulation and reuse attention maps from the source branch to maintain layout consistency (Hertz et al., 2023). Plug-and-Play extends this idea beyond attention by swapping intermediate diffusion features, enabling stronger fidelity and controllability without additional training (Tumanyan et al., 2023). MasaCtrl (Cao et al., 2023) edits images by replacing the Value features in self-attention blocks with those extracted from a reconstruction pass, thereby preserving structural consistency.Additionally, Add-it (Tewel et al., 2025) conditions the editing trajectory by injecting the source image's self-attention Key and Value features, thereby improving controllability while maintaining contextual consistency.With respect to FLUX. models, RF-Editing (Wang et al., 2025) performs reconstruction-based editing by reusing the Value features of the multimodal diffusion transformer recorded during inversion, thereby improving structural fidelity. In contrast, RF-Inversion (Rout et al., 2025) treats the original image as an explicit prior to guide both the forward and re-

verse dynamics for semantic manipulation. FireFlow (Deng et al., 2025) proposes a low-step rectified-flow inversion solver, enabling efficient reconstruction-based editing.

Beyond general semantic image editing, visual text editing and generation impose stricter requirements on glyph structure, text readability, and style preservation. Recent works have explored diffusion-based scene text editing with explicit prior guidance (Zeng et al., 2024), provided broader surveys and benchmarks for visual text processing (Shu et al., 2025), and studied style-conditioned multilingual scene text generation (Chen et al., 2026). These applications further highlight the importance of faithful inversion and structure-preserving editing.

# 3. Preliminaries

## 3.1. Rectified Flow

Rectified Flow (RF) (Liu et al., 2023) conceptualizes generative modeling as a transport problem, learning a continuous bijection between a source noise distribution $\pi_0$ (typically $\mathcal{N}(\mathbf{0}, \mathbf{I})$) and a target data distribution $\pi_1$. Unlike diffusion models that rely on stochastic dynamics, RF establishes a deterministic Ordinary Differential Equation (ODE) framework.

**Training Objective.** During training, RF constructs linear interpolation paths between sampled data pairs $(\mathbf{x}_0, \mathbf{x}_1) \sim \pi_0 \times \pi_1$. We define the interpolation trajectory as $\mathbf{x}_t = t\mathbf{x}_1 + (1 - t)\mathbf{x}_0$ for $t \in [0, 1]$. The model learns a time-dependent velocity field $v_\theta(\mathbf{z}, t)$ to approximate the drift of these paths by minimizing the flow matching objective:

$$\mathcal{L}_{\mathrm{RF}}(\theta) = \mathbb{E}_{t, \mathbf{x}_0, \mathbf{x}_1} \left[ \|v_\theta(\mathbf{x}_t, t) - (\mathbf{x}_1 - \mathbf{x}_0)\|_2^2 \right]. \quad (1)$$

By optimizing this loss, $v_\theta$ captures the marginal vector field that transports the probability density from $\pi_0$ to $\pi_1$.

**Forward Process.** In the inference phase, generation is achieved by simulating the flow forward in time. Starting from a random noise sample $\mathbf{z}_0 \sim \pi_0$, the data sample $\mathbf{z}_1$ is obtained by integrating the learned velocity field:

$$\mathbf{z}_1 = \mathbf{z}_0 + \int_0^1 v_\theta(\mathbf{z}_t, t) \, \mathrm{d}t. \quad (2)$$

This *forward process* effectively pushes the mass of the noise distribution along the learned trajectories to match the data distribution.

**Reverse Process.** A key advantage of the ODE formulation is its reversibility. Given a data sample $\mathbf{z}_1 \sim \pi_1$, we can uniquely retrieve its corresponding latent representation $\mathbf{z}_0$ by integrating the flow backward in time from $t = 1$

to $t = 0$. This *reverse process* is governed by the negated velocity field:

$$\mathbf{z}_0 = \mathbf{z}_1 + \int_1^0 v_\theta(\mathbf{z}_t, t) \, \mathrm{d}t = \mathbf{z}_1 - \int_0^1 v_\theta(\mathbf{z}_t, t) \, \mathrm{d}t. \quad (3)$$

In practice, both Eq. (2) and Eq. (3) are solved using numerical solvers (e.g., Euler or Runge-Kutta methods). Ideally, this bidirectional mapping allows for lossless reconstruction $(\mathbf{z}_1 \rightarrow \mathbf{z}_0 \rightarrow \mathbf{z}_1)$, provided that the integration is precise.

## 3.2. Motivation

**The Ideal of Euclidean Linearity.** Rectified Flow (RF)(Liu et al., 2023) is designed to transport a noise distribution $\pi_0$ to a data distribution $\pi_1$ via a deterministic Ordinary Differential Equation (ODE). The core philosophy of RF is to "straighten" the generative path in the ambient Euclidean space $\mathbb{R}^D$. Ideally, given empirical observations of $X_0 \sim \pi_0, X_1 \sim \pi_1$, the flow follows a linear trajectory $\boldsymbol{X}_t = t\boldsymbol{X}_1 + (1 - t)\boldsymbol{X}_0$, which implies a constant velocity field:

$$\frac{d\boldsymbol{X}_t}{dt} = \boldsymbol{v}(\boldsymbol{X}_t, t) = \boldsymbol{X}_1 - \boldsymbol{X}_0, \quad \text{implying} \quad \frac{d^2\boldsymbol{X}_t}{dt^2} = 0. \quad (4)$$

This property is highly desirable as it theoretically enables single-step generation and minimizes discretization error $\mathcal{O}(\Delta t^2)$ for standard Euler solvers.

**Non-Zero Acceleration.** However, empirical observations consistently contradict this linearity assumption. Trained velocity fields $v_\theta^t$ exhibit significant curvature, with time-varying velocity profiles that cannot be explained by constant-velocity motion alone. This necessitates the introduction of acceleration terms or higher-order numerical solvers in recent works, as first-order integrators fail to capture the underlying trajectory dynamics. Conventionally, this deviation is modeled as a fitting error or a numerical artifact. Specifically, existing methods attempt to approximate the trajectory via a second-order Taylor expansion:

$$\boldsymbol{X}_{t+\Delta t} \approx \boldsymbol{X}_t + \boldsymbol{v}_t \Delta t + \frac{1}{2} \boldsymbol{a}_t \Delta t^2, \quad (5)$$

where the acceleration $\boldsymbol{a}_t = \frac{dv}{dt}$ is treated as a residual term to be minimized or compensated for. This perspective presents a fundamental paradox: although the model is explicitly trained to regress the constant target $\boldsymbol{X}_1 - \boldsymbol{X}_0$, it persistently yields a non-zero acceleration $\boldsymbol{a}_t \neq 0$. Attributing this solely to optimization failure is unsatisfactory given the high capacity of modern networks.

**On the Data Manifold.** The observed curvature in learned trajectories is grounded in the fundamental *Manifold Hypothesis* (Bengio et al., 2013; Fefferman et al., 2016). Rather

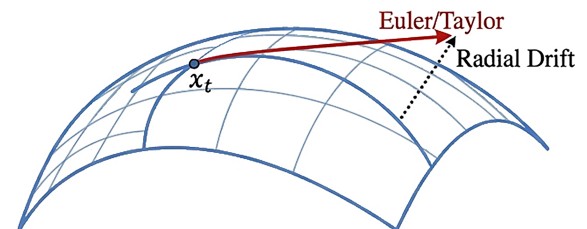
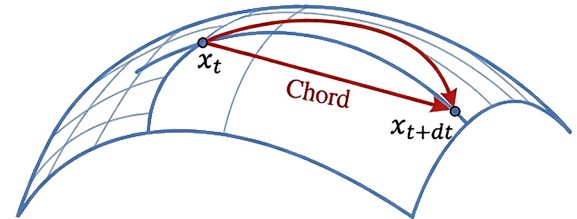

*a* **Spurious Centrifugal Drift.** Standard solvers (Euler/Taylor) move tangentially, inevitably leaving the curved manifold $\mathcal{M}$.

*b* **Chordal Update (Ours).** SlerpFlow constructs a secant vector to maintain the trajectory on the manifold shell.

*Figure 1.* **Geometric interpretation of discretization errors.** We illustrate why standard solvers fail on curved manifolds (a) and how our proposed Decoupled Chordal Update (b) eliminates the radial drift by strictly adhering to the intrinsic geometry.

than being a numerical artifact, this curvature emerges as a geometric necessity. Given that natural data resides on a low-dimensional manifold $\mathcal{M} \subset \mathbb{R}^D$—and that velocity fields parameterized by smooth neural networks naturally preserve this topological structure (Lei et al., 2020)—a strictly Euclidean straight line (Eq. 4) inevitably traverses low-density regions.

This geometric constraint is visually corroborated in Figure 1. While the learned flow naturally aligns with the tangent space of $\mathcal{M}$, the Euclidean linear assumption (Figure 1a) projects the trajectory to an off-manifold point. The deviation labeled as "Error" stems from neglecting the manifold's intrinsic curvature. To eliminate this deviation and constrain the trajectory to $\mathcal{M}$, the flow is modeled as approximating a geodesic—the straightest path intrinsically, which appears curved in the ambient space. Mathematically, the ambient acceleration $\ddot{X}_t$ decomposes into tangential and normal components:

$$\ddot{X}_t = \nabla_{\dot{X}}\dot{X} + \Pi_{\perp}(\dot{X}, \dot{X}) \tag{6}$$

where $\nabla_{\dot{X}}\dot{X}$ represents the intrinsic acceleration, and $\Pi_{\perp}(\dot{X}, \dot{X})$ denotes the extrinsic curvature.

For a geodesic, the intrinsic acceleration vanishes ($\nabla_{\dot{X}}\dot{X} = 0$). However, the extrinsic curvature term $\Pi_{\perp}$ (related to the Second Fundamental Form) remains non-zero. Thus, the "acceleration" observed in Rectified Flow corresponds to the centripetal force necessary to navigate the data manifold, establishing $a_t$ as an intrinsic geometric feature.

**Geometric Analysis of Second-Order Discretization.** Recent works (Wang et al., 2025) have empirically observed that increasing the solver order to $n = 2$ significantly mitigates these errors. We provide a geometric interpretation for this effectiveness. Expanding the trajectory to the second order yields:

$$Z_{t+h} \approx Z_t + h \cdot v_{\theta}(Z_t, t) + \frac{h^2}{2} \cdot \dot{v}_{\theta}(Z_t, t). \tag{7}$$

Here, the total time derivative $\dot{v}_{\theta}$ represents the acceleration or local curvature of the trajectory.

Geometrically, while $n = 1$ fits a tangent line, $n = 2$ fits a **parabola** that osculates the true trajectory. The inclusion of the second-order term $\frac{h^2}{2}\dot{v}$ effectively provides a "centripetal" correction that bends the step towards the manifold, compensating for the linear drift of the Euler method. This explains why $n = 2$ solvers are the current standard for Rectified Flow: they explicitly account for the non-zero curvature of the latent evolution.

## 4. Methodology

In this section, we first analyze the geometric error induced by standard Euclidean solvers. We then introduce **SlerpFlow**, a solver designed to eliminate this error via a decoupled chordal update mechanism. Finally, we present an asymmetric solver strategy to reconcile reconstruction fidelity with semantic editability.

### 4.1. Geometric Analysis: Spurious Centrifugal Drift

Standard explicit solvers (e.g., Euler) approximate the integral using discrete Euclidean steps: $Z_{t+h} = Z_t + hv_t$. Let us decompose the velocity $v_t$ into a radial component $v_{\parallel}$ and a tangential component $v_{\perp}$ with respect to the current state $Z_t$. The squared norm of the updated state is:

$$\|Z_{t+h}\|^2 = \|Z_t\|^2 + 2h\langle Z_t, v_{\parallel}\rangle + h^2(\|v_{\parallel}\|^2 + \|v_{\perp}\|^2). \tag{8}$$

The term $2h\langle Z_t, v_{\parallel}\rangle$ represents the legitimate first-order radial evolution. However, the term $h^2\|v_{\perp}\|^2$ introduces a strictly positive energy gain caused solely by linearizing the angular motion.

**Definition 4.1** (**Spurious Centrifugal Drift**). We define the quadratic term $\Delta\mathcal{E}_{drift} = h^2\|v_{\perp}\|^2$ as the *Spurious Centrifugal Drift*. It causes the trajectory to spiral outwards from the ideal manifold shell, accumulating an error of $\mathcal{O}(h)$ globally even in the absence of radial dynamics.

### 4.2. SlerpFlow: Discrete Chordal Approximation

To address this, SlerpFlow decouples the update into radial and angular components, constructing a **chordal velocity**

*Table 1.* Quantitative reconstruction comparison on PIE-Bench.

| Method | Unconditional | | | Conditional | | |
|---|---|---|---|---|---|---|
| | PSNR↑ | SSIM↑ $\times 10^2$ | LPIPS↓ $\times 10^2$ | PSNR↑ | SSIM↑ $\times 10^2$ | LPIPS↓ $\times 10^2$ |
| Euler | 11.10 | 40.51 | 43.13 | 11.43 | 37.64 | 44.31 |
| Heun | 11.77 | 42.10 | 39.96 | 12.17 | 40.17 | 41.18 |
| RF-Solver | 16.97 | 57.17 | 31.75 | 16.38 | 55.64 | 32.43 |
| FireFlow | 20.19 | 74.47 | 31.09 | 20.39 | 63.85 | 27.96 |
| **Ours** | **20.73** | **75.91** | **30.56** | **21.19** | **79.35** | **26.01** |

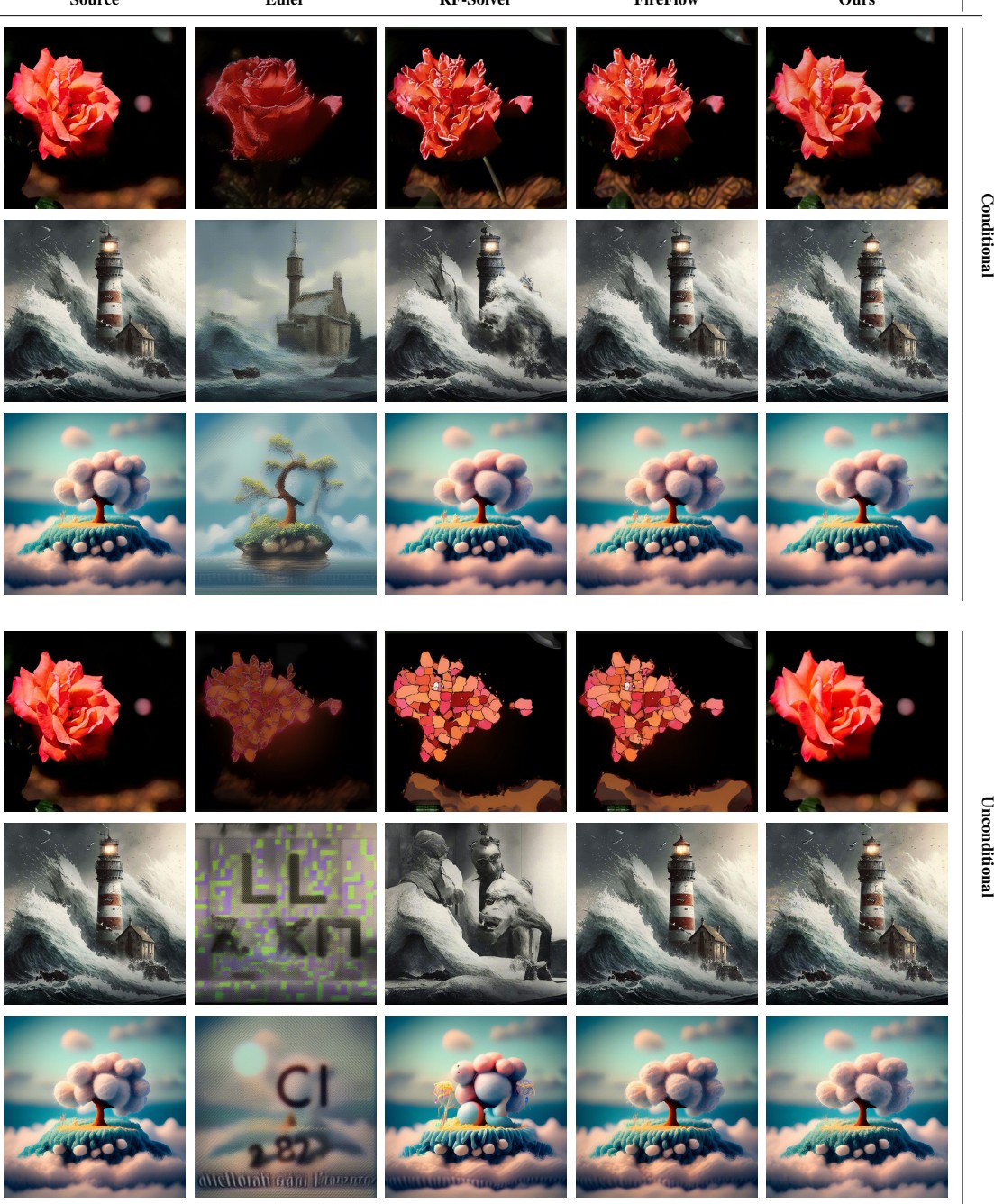

*Figure 2.* Qualitative results of image reconstruction.

that strictly adheres to the manifold geometry. Our method follows a Predictor-Corrector structure (similar to Heun's method) but operates in spherical coordinates.

**1.Radial-Angular Decoupling.** We express the latent state in polar form: $Z_t = \rho_t \cdot \mathbf{d}_t$, where $\rho_t = \|Z_t\|$ and $\mathbf{d}_t = Z_t/\rho_t \in \mathbb{S}^{d-1}$. The radial dynamics $\dot{\rho}$ and angular dynamics $\dot{\mathbf{d}}$ are treated separately.

**2.Target State Estimation.** Given the current velocity $v_t = v_\theta(Z_t, t)$ and a predicted future velocity $\hat{v}_{t+h}$ (obtained via a standard Euler predictor step), we compute the target state parameters:

- **Target Radius ($\rho_{next}$):** The radial evolution is scalar and typically smooth. We approximate it linearly (or via the learned field's radial component):

$$\rho_{next} = \rho_t + h \cdot \frac{\langle Z_t, v_t \rangle}{\|Z_t\|}. \tag{9}$$

- **Target Direction ($\mathbf{d}_{next}$):** Instead of Euclidean addition, we adhere to the intrinsic geometry. We compute the target direction using the geodesic interpolation operator $*_\alpha$:

$$\mathbf{d}_{next} = \mathbf{d}_t *_\alpha \mathbf{d}_{pred}, \tag{10}$$

where the operator is defined as $\mathbf{u} *_\alpha \mathbf{v} := \frac{\sin((1-\alpha)\Omega)}{\sin\Omega}\mathbf{u} + \frac{\sin(\alpha\Omega)}{\sin\Omega}\mathbf{v}$, with $\Omega = \arccos(\langle \mathbf{u}, \mathbf{v} \rangle)$. This formulation ensures the update traces the geodesic arc with constant angular velocity, strictly avoiding the centrifugal drift inherent to polynomial extrapolation.

**3.Chordal Velocity Synthesis.** Finally, we construct the effective update vector $v_{\text{slerp}}$ that connects the current state $Z_t$ to the geometrically corrected target $Z_{t+h}^* = \rho_{next}\mathbf{d}_{next}$:

$$v_{\text{slerp}} = \frac{Z_{t+h}^* - Z_t}{h} = \frac{\rho_{next}\mathbf{d}_{next} - Z_t}{h}. \tag{11}$$

The final update is performed as a standard linear step: $Z_{t+1} = Z_t + h \cdot v_{\text{slerp}}$.

### 4.3. Image Semantic Editing

To ensure simplicity and a fair comparison with the other solver methods, we follow the approach in (Wang et al., 2025). Similar to (Deng et al., 2025), a reference prompt is used as guidance to achieve semantic editing. Leveraging the stable inversion trajectory, our method also does not require a careful selection of timesteps for applying the replacements. Additionally, we handle the case where the angle approaches zero. The inversion and denoising sampling processes are detailed in Algorithm 1.

## 5. Experiment

### 5.1. Implementation Details

**Baselines.** To rigorously demonstrate the effectiveness of our proposed method, we benchmark against a comprehensive set of fundamental and state-of-the-art RF-based solvers.

- **Inversion & Reconstruction:** We select four representative solvers: the fundamental Euler and Heun methods, alongside the advanced RF-Solver (Wang et al., 2025) and FireFlow (Deng et al., 2025). We conduct a direct comparison between these inversion-based baselines and our method under the same experimental settings.

- **Editing:** We consider two families of inversion-based editing baselines. For diffusion-model inversion-based editing, we include P2P(Hertz et al., 2023), PnP(Tumanyan et al., 2023), PnP-Inv(Ju et al., 2024), MasaCtrl(Cao et al., 2023), and InfEdit(Xu et al., 2024).
  We further include recent rectified-flow inversion methods, RF-Inversion (Rout et al., 2025),RF-Solver (Wang et al., 2025)and FireFlow(Deng et al., 2025). We conduct a direct comparison between these editing baselines and our method under the same experimental settings.

**Implementation Details.** In experiments, we employ Flux.1-dev (Black Forest Labs, 2024) as the underlying pre-trained model for both reconstruction and editing tasks. To ensure a fair and rigorous comparison, we strictly control the computational budget:

- **For Inversion/Reconstruction:** For FLUX reconstruction, we use FireFlow with 15 sampling steps as the reference computational budget. Since different solvers require different numbers of model evaluations per sampling step, we follow the corresponding step settings to keep the NFE comparable to this reference budget. Euler, Heun, and RF-Solver baseline results follow the FLUX reconstruction setting reported by UniEdit-Flow (Jiao et al., 2026), while FireFlow and SlerpFlow are evaluated under our corrected 15-step setting.

- **For Editing:** We adopt the optimal settings reported in the original papers for FireFlow, RF-Solver, and RF-Inversion. To provide a comprehensive evaluation, we also compare our method with several diffusion-based editing approaches, including P2P(Hertz et al., 2023), PnP(Tumanyan et al., 2023), PnP-Inv(Ju et al., 2024), MasaCtrl(Cao et al., 2023), and InfEdit(Xu et al., 2024)

*Table 2.* Text-driven image editing comparison on PIE-Bench.

| Method | Model | Structure Distance↓×10³ | BG Preservation | | CLIP Sim.↑ | | Steps | NFE |
|---|---|---|---|---|---|---|---|---|
| | | | PSNR↑ | SSIM↑×10² | Whole | Edited | | |
| P2P | Diffusion | 69.43 | 17.87 | 71.14 | 25.01 | 22.44 | 50 | 100 |
| PnP | Diffusion | 28.22 | 22.28 | 79.05 | 25.41 | 22.55 | 50 | 100 |
| PnP-Inv. | Diffusion | 24.29 | 22.46 | 79.68 | 25.41 | 22.62 | 50 | 100 |
| MasaCtrl | Diffusion | 28.38 | 22.17 | 79.67 | 23.96 | 21.16 | 50 | 100 |
| InfEdit | Diffusion | 13.78 | 28.51 | 85.66 | 25.03 | 22.22 | 12 | 72 |
| RF-Inv. | FLUX | 40.60 | 20.82 | 71.92 | 25.20 | 22.11 | 28 | 56 |
| RF-Solver | FLUX | 25.53 | 21.91 | 87.51 | 26.00 | 22.88 | 15 | 60 |
| FireFlow | FLUX | 22.05 | 22.47 | 89.53 | 26.02 | 25.96 | 15 | 32 |
| **Ours** | FLUX | 23.35 | 21.14 | 88.21 | **26.79** | **26.38** | 15 | 32 |

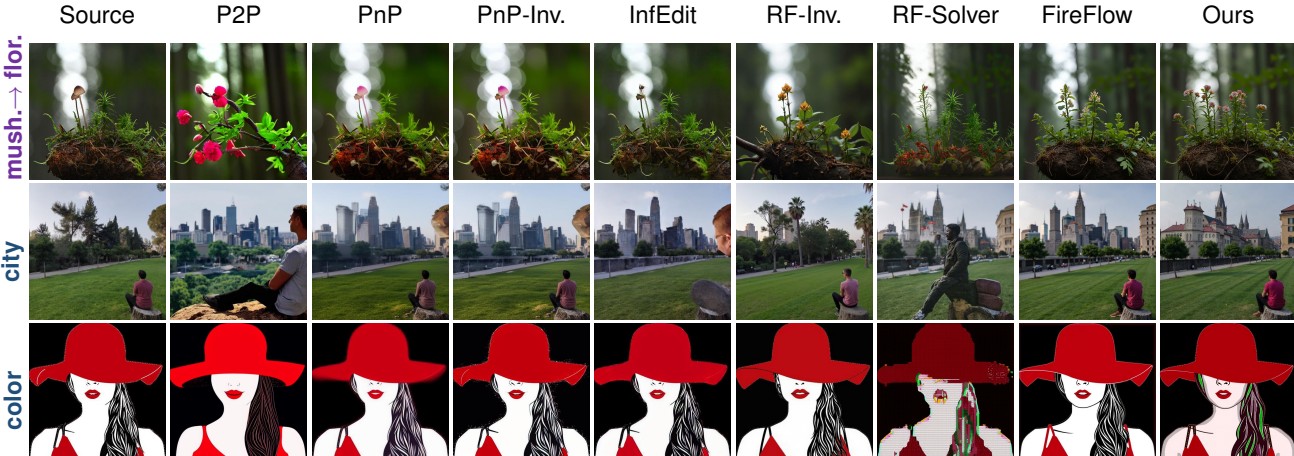

*Figure 3.* **Qualitative comparison on image editing.**

**Evaluation Metrics.** All quantitative experiments are conducted on the PIE-Bench dataset (Ju et al., 2024). We employ a comprehensive set of metrics to evaluate performance across three distinct dimensions:

- **Reconstruction Quality & Background Preservation:** To assess the fidelity of inversion and the preservation of unedited regions, we utilize four standard metrics: Peak Signal-to-Noise Ratio (PSNR) (Huynh-Thu & Ghanbari, 2008), Structural Similarity Index Measure (SSIM) (Wang et al., 2004), and Learned Perceptual Image Patch Similarity (LPIPS) (Zhang et al., 2018). For the inversion task, these metrics are calculated over the entire image to measure reconstruction accuracy. For the editing task, they are computed exclusively within the unedited background regions (using the provided masks) to evaluate the solver's ability to prevent unwanted changes.

- **Editing:** To measure how well the edited content aligns with the text prompt, we employ the CLIP Similarity score (Radford et al., 2021). We report both CLIP-Whole (similarity between the prompt and the entire image) and CLIP-Edit (similarity between the prompt and the cropped edited region) to capture global and local semantic consistency.

- **Structural Integrity:** Following Ju et al. (2024), we employ the Structure Distance metric. This evaluates the preservation of edit-irrelevant structural layout, ensuring that the geometric composition of the image remains stable during the generative editing process.

**Hyperparameters.** Unless otherwise specified, all hyperparameters are kept consistent with FireFlow for fair comparison. The only SlerpFlow-specific hyperparameter is the interpolation weight $\alpha$, which controls the strength of spherical correction.

### 5.2. Inversion and Reconstruction

**Quantitative Comparison.** Table 1 reports quantitative results on PIE-Bench under matched computational budgets. Across both conditional and unconditional settings, our method consistently outperforms Euler, Heun, RF-Solver, and FireFlow across all reconstruction metrics, demonstrating improved inversion stability and reconstruction quality under the same sampling budget. Compared with Fire-Flow, SlerpFlow achieves higher PSNR and SSIM and lower

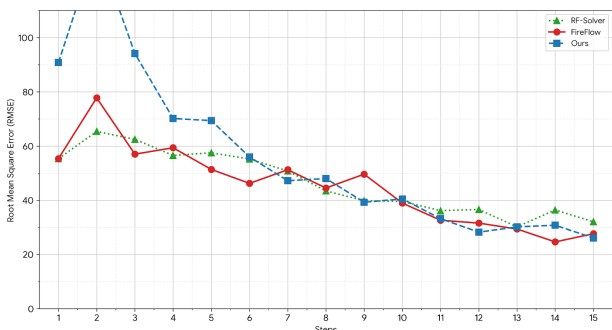

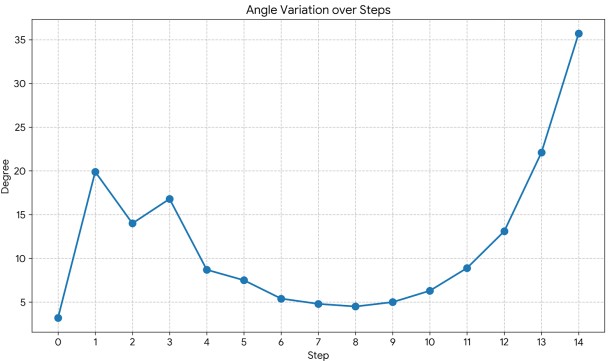

*Figure 4.* **Convergence comparison.** Reconstruction RMSE of RF-Solver, FireFlow, and SlerpFlow under different sampling steps. SlerpFlow shows decreasing error as the sampling budget increases and reaches competitive reconstruction accuracy in the middle-to-high step range.

*Figure 5.* **Angle variation of SlerpFlow.** The spherical interpolation angle used by SlerpFlow changes non-uniformly along the inversion trajectory, with larger corrections in the early and late stages and smaller corrections in the middle range. This supports step-dependent angular correction in rectified-flow inversion.

LPIPS in both conditional and unconditional settings. Relative to the Euler baseline, SlerpFlow substantially improves reconstruction fidelity, increasing PSNR and SSIM while reducing LPIPS by a large margin. These results indicate that spherical trajectory correction provides a reliable improvement for rectified-flow inversion without increasing the number of sampling steps.

**Qualitative Comparison.** As shown in Figure 2, our approach enables an efficient and effective reconstruction pipeline on Flux.1-dev. Compared to baseline solvers, it exhibits substantially less drift from the source image (e.g., fewer color/texture shifts and structural deviations), which aligns with the quantitative gains in Table 1.

**Convergence Rate:** We compare the reconstruction error of SlerpFlow with FireFlow and RF-Solver across different sampling steps, as shown in Figure 4. The updated convergence curve shows that SlerpFlow does not dominate the baselines in the extremely low-step regime. When only a very small number of sampling steps is used, the angular correction can be more pronounced, and the resulting reconstruction error may be higher than that of FireFlow and RF-Solver. As the sampling budget increases, however, the reconstruction error of SlerpFlow decreases steadily and becomes comparable to the competing solvers in the middle-to-high step range. At 15 steps, SlerpFlow achieves competitive reconstruction accuracy without requiring additional model evaluations per step. This behavior suggests that SlerpFlow is better understood as a geometric regularization mechanism for practical inversion budgets rather than an extreme few-step acceleration method.

To further analyze this behavior, we visualize the spherical interpolation angle used by SlerpFlow across sampling steps in Figure 5. The angle varies non-uniformly along the inversion trajectory: it is relatively large in the early stage,

decreases in the middle range, and increases again near the late stage. This indicates that the required geometric correction is step-dependent rather than constant. The larger angles in the early and late stages suggest stronger local angular adjustment, while the smaller angles in the middle range indicate a smoother segment of the trajectory. Together, the convergence curve and angular analysis support our view that rectified-flow inversion benefits from step-dependent spherical correction, especially under moderate sampling budgets.

### 5.3. Editing

**Quantitative Comparison.** We evaluate prompt-guided editing on PIE-Bench (Ju et al., 2024). Table 2 reports preservation metrics (PSNR/SSIM and LPIPS/MSE computed on non-edited regions), editability metrics (CLIP similarity on the whole image and the edited region), and efficiency (NFE). SlerpFlow obtains the highest CLIP-Whole and CLIP-Edited scores among all compared methods, indicating stronger semantic alignment with the target prompt. At the same time, its background preservation remains competitive under the same steps and NFE as FireFlow, although FireFlow and InfEdit achieve better scores on some preservation metrics. This behavior reflects a trade-off introduced by the global angular correction of SlerpFlow: stronger prompt-driven semantic alignment may slightly perturb unedited background regions.

**Qualitative Comparison.** Compared with baseline methods, SlerpFlow produces target-aligned edits that more strongly reflect the input prompt. While some baselines preserve the background slightly better in certain cases, they often show weaker semantic changes. Overall, the qualitative results are consistent with Table 2: SlerpFlow favors stronger semantic alignment while maintaining reasonable

source-image consistency.

## 6. Conclusion

In this work, we revisited rectified-flow inversion from a geometric perspective and identified spurious centrifugal drift as a source of discretization error in Euclidean solver updates. Motivated by the hyperspherical structure of high-dimensional latent spaces, we introduced SlerpFlow, a training-free geometric correction method that decouples radial and angular components and applies spherical interpolation to regularize the inversion trajectory. Our corrected 15-step evaluation shows that SlerpFlow consistently improves reconstruction fidelity over Euler, Heun, RF-Solver, and FireFlow without requiring additional model evaluations. For text-guided editing, SlerpFlow achieves stronger CLIP-based semantic alignment, while its background preservation remains competitive but slightly weaker than FireFlow on some metrics. These results suggest that SlerpFlow is best understood as a practical geometric regularizer for rectified-flow inversion rather than an extreme few-step acceleration method.

**Limitations and Future Work.** Despite these advantages, SlerpFlow relies on an approximate hyperspherical latent-space assumption and may introduce over-correction when the local manifold geometry is highly anisotropic or locally flat. In editing, the global angular correction can also perturb unedited background regions when the target prompt induces a strong semantic shift. Future work may explore adaptive curvature estimation, region-aware editing corrections, and training-time spherical interpolation objectives that align the learned flow more directly with the curved latent geometry.

## Acknowledgements

This work was supported by the Beijing Natural Science Foundation (Grant number: 4262075), Research Funds for NSD Construction, University of International Relations (Grant numbers: 3262026T23).

## Impact Statement

This paper presents work whose goal is to advance the field of Machine Learning. There are many potential societal consequences of our work, none which we feel must be specifically highlighted here.

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

## A. Algorithm

---

**Algorithm 1** SlerpFlow

---

1: **Input:** Current state $Z_t$, model $v_\theta$, step size $h$.
2: **1. Predictor (Euler):**
3:     $\hat{v}_t \leftarrow v_\theta(Z_t, t)$
4:     $\hat{Z}_{t+h} \leftarrow Z_t + h \cdot \hat{v}_t$
5: **2. Corrector (Geometry-Aware):**
6:     $\hat{v}_{t+h} \leftarrow v_\theta(\hat{Z}_{t+h}, t+h)$
7:     $\bar{v} \leftarrow (\hat{v}_t + \hat{v}_{t+h})/2$ {Average Velocity}
8: **3. Geometric Decomposition:**
9:     $\rho_t \leftarrow \|Z_t\|$
10:     $\rho_{next} \leftarrow \rho_t + h \cdot \langle Z_t/\rho_t, \bar{v} \rangle$ {Target Radius}
11:     $\mathbf{d}_t \leftarrow Z_t/\rho_t$
12:     $\mathbf{d}_{euler} \leftarrow (Z_t + h\bar{v})/\|Z_t + h\bar{v}\|$
13:     $\mathbf{d}_{next} \leftarrow \mathrm{Slerp}(\mathbf{d}_t, \mathbf{d}_{euler}, 0.5)$ {Corrected Direction}
14: **4. Chordal Update:**
15:     $v_{\mathrm{slerp}} \leftarrow (\rho_{next}\mathbf{d}_{next} - Z_t)/h$
16:     $Z_{t+1} \leftarrow Z_t + h \cdot v_{\mathrm{slerp}}$
17: **Return** $Z_{t+1}$

---

**NFE accounting and velocity caching.** Algorithm 1 describes a single SlerpFlow update for clarity. In our implementation, the velocity at the current state is cached from the previous step and reused in the next step. Therefore, after the first initialization step, only the future velocity $\hat{v}_{t+h}$ requires a new model evaluation. This caching strategy keeps the number of function evaluations comparable to FireFlow. For editing, both inversion and denoising trajectories are executed with the same step budget; thus, with 15 steps, the total NFE is 32, matching the setting reported in Table 2.

## B. Mathematical Proofs

### B.1. Proof of Manifold Deviation

Consider a trajectory on a hypersphere $\mathbb{S}^{D-1}$ of radius $R$. Let $Z_t$ be the state with $\|Z_t\| = R$. For a standard Euclidean Euler step $Z_{Euler} = Z_t + hv_t$, even if the velocity $v_t$ is strictly tangential ($\langle Z_t, v_t \rangle = 0$):

$$\|Z_{Euler}\|^2 = \|Z_t\|^2 + 2h\langle Z_t, v_t \rangle + h^2\|v_t\|^2 = R^2 + h^2\|v_t\|^2. \tag{12}$$

Applying Taylor expansion $\|Z_{Euler}\| \approx R + \frac{h^2\|v_t\|^2}{2R}$. This proves that Euclidean solvers introduce a systematic radial drift of order $\mathcal{O}(h^2)$. SlerpFlow eliminates this drift by explicitly projecting the state back to the predicted manifold shell.

### B.2. Local Truncation Error (LTE) Derivation

Let $Z(t)$ be the exact solution with expansion $Z_{true}(t+h) = Z(t) + hv(t) + \frac{h^2}{2}\dot{v}(t) + \mathcal{O}(h^3)$. SlerpFlow's effective velocity $v_{slerp}$ linearizes to the trapezoidal mean:

$$v_{slerp} \approx \frac{v(t) + v(t+h)}{2} = v(t) + \frac{h}{2}\dot{v}(t) + \mathcal{O}(h^2). \tag{13}$$

Substituting into the update $Z_{slerp} = Z_t + hv_{slerp}$:

$$Z_{slerp}(t+h) = Z(t) + hv(t) + \frac{h^2}{2}\dot{v}(t) + \mathcal{O}(h^3). \tag{14}$$

Thus, SlerpFlow matches the exact solution up to $h^2$, yielding an LTE of $\mathcal{O}(h^3)$ and a global convergence order of $\mathcal{O}(h^2)$.

## C. Theoretical Analysis

In this section, we provide rigorous mathematical derivations to support the geometric motivation (Section 3) and the convergence properties of the proposed method (Section 4).

### C.1. Geometric Analysis of Spurious Centrifugal Drift

In the Motivation section, we argue that standard Euclidean solvers introduce a systematic energy inflation, pushing trajectories off the data manifold. Here, we quantify this error.

**Assumption (Locally Spherical Dynamics).** For clarity of derivation, we consider a latent state $Z_t$ constrained to a hypersphere of radius $R$ (i.e., $||Z_t|| = R$). We assume the learned velocity field $v_t$ describes a pure rotation locally, meaning it is strictly tangent to the manifold:

$$\langle Z_t, v_t \rangle = 0. \tag{15}$$

While real-world generative flows may contain legitimate radial evolution, the derivation below isolates the *spurious* drift component introduced solely by the discretization of the tangential dynamics.

**Derivation.** Consider a standard Euclidean Euler update with step size $h$: $Z_{next} = Z_t + hv_t$. The squared norm of the updated state is:

$$||Z_{next}||^2 = ||Z_t + hv_t||^2 = ||Z_t||^2 + 2h\langle Z_t, v_t \rangle + h^2||v_t||^2. \tag{16}$$

Applying the tangential assumption ($\langle Z_t, v_t \rangle = 0$) and substituting $||Z_t|| = R$:

$$||Z_{next}||^2 = R^2 + h^2||v_t||^2 = R^2 \left( 1 + \frac{h^2||v_t||^2}{R^2} \right). \tag{17}$$

Taking the square root and applying the Taylor expansion $\sqrt{1+x} \approx 1 + x/2$ for small $h$:

$$||Z_{next}|| \approx R \left( 1 + \frac{h^2||v_t||^2}{2R^2} \right) = R + \frac{h^2||v_t||^2}{2R}. \tag{18}$$

**Conclusion.** Equation (18) reveals a strictly positive radial error term $\Delta R \approx \frac{h^2||v_t||^2}{2R}$. This confirms that Euclidean solvers introduce an $\mathcal{O}(h^2)$ **Spurious Centrifugal Drift** at every step, causing the trajectory to spiral outwards from the high-density manifold shell.

### C.2. Convergence and Accuracy Analysis

Here, we demonstrate that SlerpFlow achieves second-order convergence, justifying its improved reconstruction fidelity compared to first-order Euler methods.

**Exact Solution Expansion.** Let $Z(t)$ be the exact solution to the ODE $dZ_t = v(Z_t, t)dt$. Assuming the velocity field is differentiable, the Taylor expansion of the true state at $t + h$ is:

$$Z_{true}(t + h) = Z(t) + hv(t) + \frac{h^2}{2}\dot{v}(t) + \mathcal{O}(h^3), \tag{19}$$

where $\dot{v}(t)$ denotes the total time derivative (acceleration).

**SlerpFlow Update Analysis.** SlerpFlow constructs an effective chordal velocity $v_{slerp}$ to update the state: $Z_{slerp} = Z_t + h \cdot v_{slerp}$. The core of SlerpFlow utilizes Spherical Linear Interpolation (Slerp) with a factor $\alpha = 0.5$ between the current direction $d_t$ and the predicted direction $d_{pred}$. In the limit of small step size $h$, the geodesic chord constructed by Slerp asymptotically aligns with the arithmetic mean of the tangent vectors at the endpoints (trapezoidal integration):

$$v_{slerp} \approx \frac{v(t) + v(t + h)}{2}. \tag{20}$$

Substituting the expansion $v(t + h) \approx v(t) + h\dot{v}(t)$ into the above:

$$v_{slerp} \approx v(t) + \frac{h}{2}\dot{v}(t) + \mathcal{O}(h^2). \tag{21}$$

Now, substituting this effective velocity into the linear update step:

$$Z_{slerp}(t + h) = Z(t) + h\left(v(t) + \frac{h}{2}\dot{v}(t)\right) = Z(t) + hv(t) + \frac{h^2}{2}\dot{v}(t) + \mathcal{O}(h^3). \tag{22}$$

**Conclusion.** Comparing Equation (22) with the exact expansion in Equation (19), we observe that SlerpFlow matches the exact solution up to the $h^2$ term. Consequently, the **Local Truncation Error (LTE)** is $\mathcal{O}(h^3)$, which implies a **Global Convergence Order** of $\mathcal{O}(h^2)$. This theoretically validates why SlerpFlow outperforms Euler methods and achieves accuracy comparable to higher-order solvers like Heun, while strictly adhering to the manifold geometry.

**Small Angles.** The operator $*_\alpha$ defined in Eq. 10 relies on spherical linear interpolation (Slerp). A numerical instability arises when the angle $\Omega = \arccos(\langle \mathbf{u}, \mathbf{v} \rangle)$ approaches 0, causing the denominator $\sin\Omega$ to vanish.

To address this, we apply the small-angle approximation $\lim_{\theta \to 0} \sin\theta \approx \theta$. When $\Omega < \epsilon$ (where $\epsilon$ is a small threshold, e.g., $10^{-6}$), the operator degrades to linear interpolation (Lerp):

$$\lim_{\Omega \to 0}\left(\frac{\sin((1-\alpha)\Omega)}{\sin\Omega}\mathbf{u} + \frac{\sin(\alpha\Omega)}{\sin\Omega}\mathbf{v}\right) = (1 - \alpha)\mathbf{u} + \alpha\mathbf{v}. \tag{23}$$

This linear approximation ensures gradients remain bounded and prevents numerical divergence during the identity mapping or fine-grained editing tasks.

## D. Ablation Study

### D.1. Reconstruction comparison in SD3.

*Table 3.* Quantitative reconstruction comparison in SD3.

| | **Steps** | **NFE↓** | **LPIPS↓** | **SSIM↑** | **PSNR↑** |
|---|---|---|---|---|---|
| DDIM-Inv. | 50 | 100 | 0.2342 | 0.5872 | 19.72 |
| ReFlow-Inv. | 30 | 60 | 0.5044 | 0.5632 | 16.57 |
| RF-Solver. | 30 | 120 | 0.2926 | 0.7078 | 20.05 |
| RF-Inversion. | 30 | 60 | 0.4480 | 0.4599 | 16.22 |
| FireFlow. | 30 | 62 | 0.2336 | 0.8024 | 24.01 |
| Ours. | 30 | 62 | **0.2121** | **0.8241** | **24.21** |

To further evaluate the transferability of our geometric prior, we extend SlerpFlow to SD3. As shown in Table 3, SlerpFlow achieves the best results among the compared methods, with modest but consistent gains over FireFlow. Although the absolute reconstruction fidelity on SD3 differs from that on FLUX, the results suggest that spherical trajectory correction can still provide useful reconstruction improvements beyond a single rectified-flow model. We attribute the inter-model performance gap to differences in the intrinsic curvature of the learned velocity fields: FLUX is optimized for straighter trajectories, whereas SD3 may exhibit stiffer dynamics with higher local curvature. In this setting, SlerpFlow remains competitive by mitigating radial drift without relying on unstable derivative estimates

## E. Limitations and Future Perspectives

### E.1. Theoretical Limitations

**Homogeneity of Curvature Assumption.** SlerpFlow mitigates centrifugal drift by leveraging the empirical prior that high-dimensional latent representations often concentrate near thin shells, where local geometry can be approximated by a

hypersphere. This implicitly assumes a *nearly constant* local curvature. When the underlying manifold exhibits strongly varying sectional curvature, or becomes locally flatter in certain regions, enforcing spherical interpolation may lead to mild over-correction, potentially introducing subtle artifacts in rare cases.

**Linear Radial Approximation.**    Our update decouples the state into radial and angular components, $Z_t = \rho_t \cdot d_t$, and models the radial evolution with a linear approximation. While this is sufficient under typical step sizes, it may underfit rapid energy fluctuations during highly non-linear phases of Rectified Flow dynamics. In principle, a fully coupled high-order solver could better capture such transitions, at the cost of increased computation and sensitivity to model noise.

### E.2. Future Perspective: A Lie Group View

**Towards Lie-Group Geometric Integrators.**    A promising direction is to generalize the *angular* update of SlerpFlow through Lie-group methods. Our Slerp step is a geodesic update on the unit sphere $\mathbb{S}^{D-1}$; equivalently, it can be realized via the action of a minimal rotation $R \in SO(D)$ that maps the current direction to a target direction, i.e., $d_{t+h} = R^\alpha d_t$ for some step fraction $\alpha$. This viewpoint suggests constructing exponential-map-based updates, where one parameterizes a local generator $\Omega \in \mathfrak{so}(D)$ and applies $R = \exp(\Omega)$ to obtain intrinsically structure-preserving angular transport. More broadly, formulating the transport on homogeneous spaces or symmetry-structured manifolds may enable solvers that better respect intrinsic invariants (e.g., norm and group-action constraints), potentially improving stability and intrinsic accuracy with reduced reliance on ad-hoc projection heuristics. We leave a rigorous development—including how to infer $\Omega$ from learned velocities under model noise—to future work.

