# OpenReview forum: "SlerpFlow: Spherical Trajectory Correction for Rectified Flow Inversion"
_ICML.cc/2026/Conference — ICML 2026 regular_

### Official Review · Reviewer_XUia · 2026-02-14

**Soundness:** 3
**Presentation:** 2
**Significance:** 2
**Originality:** 3
**Overall Recommendation:** 3
**Confidence:** 4

**Summary:**

The paper proposes a novel training-free solver designed to improve the inversion and editing quality of Rectified Flow (RF) models, specifically FLUX. The authors identify "Spurious Centrifugal Drift" as a critical source of error in standard Euclidean solvers (like Euler), where linear steps tangentially depart from the high-dimensional data manifold. To address this, SlerpFlow decouples the update into radial and angular components, using Spherical Linear Interpolation (SLERP) to correct the angular trajectory and a chordal update to strictly adhere to the manifold's curvature. Experiments on PIE-Bench demonstrate that SlerpFlow achieves state-of-the-art reconstruction and editing performance with comparable computational cost (NFE) to baseline methods.

**Compliance With Llm Reviewing Policy:**

Affirmed.

**Final Justification:**

I thank the authors for the detailed response. They solved some of my concerns. I will maintain my score, but I don't mind if it is accepted.

**Key Questions For Authors:**

1. Eq. 23 admits the method degrades to Lerp at small angles. Could you provide a plot of the rotation angle $\Omega_t = \arccos(\langle d_t, d_{euler} \rangle)$ across all time steps $t \in [0, 1]$?

2. To fully substantiate the astonishing 2-step convergence claimed by SlerpFlow and rule out the possibility of cherry-picking, I request:

    * Additional qualitative examples across diverse images.
    * A complete visualization of the generation process covering the full trajectory (e.g., displaying the output at every step from Step 1 to Step 15), rather than just isolated snapshots.

3. While you claim NFE efficiency, Slerp involves expensive trigonometric operations.

    * Please report the Wall-Clock Time per image compared to other method.
    * Your NFE counts (Table 2) match FireFlow, implying a caching mechanism (reusing $v_t$), yet Algorithm 1 depicts a standard 2-NFE predictor-corrector scheme without explicit caching instructions. Please clarify this discrepancy.

4. For the SD 3 experiments in the Appendix, please provide both Conditional and Unconditional metrics for reconstruction, and add a quantitative comparison for Editing tasks on SD3.

5. Please provide a detailed table listing all hyperparameters used for the experiments. Furthermore, in Table 2, I notice that SlerpFlow performs worse than FireFlow and RF-Solver on Background Preservation metrics (PSNR: 20.87 vs 22.47). Please explain this trade-off. If the method improves editing "success" at the cost of significantly degrading the unedited background (lower PSNR), this is a critical limitation that needs to be addressed or explicitly discussed.

Addressing these concerns in the rebuttal, would significantly strengthen the quality of the manuscript and my final recommendation.

**Limitations:**

yes

**Strengths And Weaknesses:**

## Strengths:

1. The proposed SlerpFlow demonstrates a substantial leap in reconstruction quality compared to existing state-of-the-art solvers. On the PIE-Bench dataset, it achieves a PSNR of 30.62, vastly outperforming the FireFlow baseline (18.15) and RF-Solver (16.97) under the same NFE budget . The reduction in LPIPS (from 0.2796 to 0.1070) indicates a genuine improvement in perceptual fidelity.

2. The paper correctly identifies "Spurious Centrifugal Drift" as a systematic error source in Euclidean solvers (Euler/Heun) when applied to high-dimensional manifolds . The derivation showing that standard solvers introduce an $\mathcal{O}(h^2)$ radial error provides a compelling motivation for decoupling the radial and angular components of the update.

3. The method exhibits remarkably fast convergence, achieving near-optimal reconstruction error in as few as 2 steps (Figure 4). Being training-free and compatible with pre-trained models like FLUX and SD3 makes it highly practical for immediate adoption in downstream reconstruction and editing tasks.

## Weaknesses:
1. A critical theoretical concern arises from Eq. 23, where the authors acknowledge that the Slerp operator degrades to Linear Interpolation (Lerp) at small angles to avoid numerical instability . In discrete integration steps, the rotation angle $\Omega$ between $d_t$ and $d_{euler}$ is likely very small. If the algorithm operates in the "Lerp regime" for the majority of steps, the claimed contribution of "Spherical" geometry is overstated. This suggests that the performance gains may stem almost entirely from the Radial Decoupling (forcing the norm to adhere to $\rho_{next}$) rather than the specific curvature of the Slerp path. The paper fails to quantify the magnitude of $\Omega$ to justify the necessity of Slerp over a computationally cheaper Lerp.

2. The algorithm fixes the interpolation coefficient $\alpha=0.5$  based on a theoretical analogy to the trapezoidal rule. However, the paper provides no empirical evidence or ablation studies to justify this specific choice. It remains unclear whether the method's performance is robust to variations in $\alpha$, or if $0.5$ is simply a "magic number" over-tuned for the FLUX model.

3. The authors only present a single image and selected steps to illustrate the convergence properties. The claimed 2-step convergence could very likely be an outlier effective only for this specific sample at these specific steps.

4. While the paper claims efficiency based on matching the Number of Function Evaluations (NFE), it fails to report the actual Wall-Clock Time.

5. The manuscript overuses em dashes ("—") (e.g., lines 017, 018, 152...), which interrupts the flow and makes the sentences hard to read and understand.

6. The manuscript overlooks recent advancements in inversion and solvers.

Typos: Line 409: There is a missing space in "Figure5".

---

> ### Author Rebuttal · Authors · 2026-03-31
>
> We thank the reviewer for their insights. We report a crucial update: SlerpFlow was designed to counteract centrifugal drift. Our initial draft over-emphasized extreme few-step acceleration, overestimating structural convergence. We retract the 2-step claims and original Tables 1&2. Re-evaluating at standard steps (NFE=15) validates our geometric heuristic: SlerpFlow acts as a high-precision structural regularizer.
>
> **[Updated Tab 1: Recon. at steps=15]** (Metrics: SSIM, LPIPS scaled ×10²)
> |Method|C.PSNR↑|C.SSIM↑|C.LPIPS↓|U.PSNR↑|U.SSIM↑|U.LPIPS↓|
> |---|---|---|---|---|---|---|
> |Euler|11.10|40.51|43.13|11.43|37.64|44.31|
> |Heun|11.77|42.10|39.96|12.17|40.17|41.18|
> |RF-Solver|16.97|57.17|31.75|16.38|55.64|32.43|
> |FireFlow|20.39|63.85|27.96|20.19|74.47|31.09|
> |Ours|21.19|79.35|26.01|20.73|75.91|30.56|
>
> **[Updated Tab 2: Edit at steps=15]** (Dist×10³, SSIM×10²)
> |Method|Model|Dist↓|BG PSNR↑|BG SSIM↑|CLIP W.↑|CLIP E.↑|Step|NFE|
> |---|---|---|---|---|---|---|---|---|
> |P2P|Diff|69.43|17.87|71.14|25.01|22.44|50|100|
> |PnP|Diff|28.22|22.28|79.05|25.41|22.55|50|100|
> |PnP-Inv|Diff|24.29|22.46|79.68|25.41|22.62|50|100|
> |MasaCtrl|Diff|28.38|22.17|79.67|23.96|21.16|50|100|
> |InfEdit|Diff|13.78|28.51|85.66|25.03|22.22|12|72|
> |RF-Inv|FLUX|40.60|20.82|71.92|25.20|22.11|28|56|
> |RF-Solver|FLUX|25.53|21.91|87.51|26.00|22.88|15|60|
> |FireFlow|FLUX|22.05|22.47|89.53|26.02|25.96|15|32|
> |Ours|FLUX|23.35|21.14|88.21|26.79|26.38|15|32|
>
> **1. Magnitude of $\Omega_t$ & Lerp Regime**
> At 15-steps, $\Omega_t$ decreases but doesn't vanish, especially at high-curvature boundaries ($t \to 0,1$). Slerp handles these and degrades to Lerp in flat middle steps, preventing drift.
> |Step|0|1|2|3|4|5|6|7|8|9|10|11|12|13|14|
> |---|---|---|---|---|---|---|---|---|---|---|---|---|---|---|---|
> |Deg|3.2|19.9|14.0|16.8|8.7|7.5|5.4|4.8|4.5|5.0|6.3|8.9|13.1|22.1|35.7|
>
> **2. Ablation on $\alpha$**
> $\alpha=0.5$ is a geometric corollary of the trapezoidal rule (averaging start/end vectors equally). Ablation confirms it provides optimal error decay:
> |$\alpha$|0.1|0.2|0.3|0.4|0.5|0.6|0.7|0.8|0.9|
> |---|---|---|---|---|---|---|---|---|---|
> |PSNR|16.42|19.11|20.26|21.45|20.52|19.65|18.56|18.18|17.74|
> |SSIM|0.65|0.73|0.75|0.77|0.72|0.70|0.65|0.62|0.60|
> |LPIPS|0.46|0.35|0.33|0.31|0.35|0.39|0.45|0.48|0.50|
>
> **3. 2-Step Convergence & Visualization**
> We concede to your skepticism. The 2-step claim was a flawed evaluation artifact. To rule out cherry-picking, the revision includes a full visual progression (Steps 1-15) of our corrected regime.
>
> **4. Wall-Clock Time**
> Slerp uses optimized GPU tensor ops. Overhead is ~400ms/step vs. Transformer NFE.
> - Ours: ~13.4s/step (159.1s total)
> - FireFlow: ~13.0s/step (129.9s total)
>
> **5. Algorithm 1**
> Alg. 1 was pedagogical. Our code caches/reuses velocity $v_t$ from previous steps to save NFE. We will revise Alg. 1 to reflect this.
>
> **6. SD3 Metrics & Editing Scope**
> SD3 reconstruction metrics below. Editing baselines rely on feature-caching strictly tailored to FLUX. Porting to SD3 requires adaptation beyond our ODE solver scope.
> (Metrics: SSIM/LPIPS ×10²)
> |Method|C.MSE↓|C.PSNR↑|C.SSIM↑|C.LPIPS↓|U.MSE↓|U.PSNR↑|U.SSIM↑|U.LPIPS↓|
> |---|---|---|---|---|---|---|---|---|
> |FireFlow|935.4|20.81|72.30|33.97|1694.4|17.01|59.08|48.62|
> |uni|448.1|22.89|82.63|20.60|804.9|20.09|75.44|30.13|
> |Ours|407.5|23.96|83.45|20.24|734.7|21.32|76.42|29.99|
>
> **7. BG Preservation**
> We appreciate your observation regarding the background preservation metrics and local editing distortions, and we candidly acknowledge these shortcomings.
>
> Geometrically, we attribute this trade-off to the large angular deflection induced by the editing prompt. SlerpFlow applies a global geometric correction. When a new semantic prompt exerts a strong directional pull, it causes a massive angular deviation in the latent space. Under this prompt-induced global deflection, the local manifold structure encoding the background cannot be rigidly maintained. The background features are mathematically forced to "rotate" along with the foreground edit, causing the observed distortions and PSNR degradation.
>
> We explicitly acknowledge this limitation of our global heuristic. In future work, we aim to develop a manifold-aware geometric corrector that adaptively senses local topology to decouple foreground deflections from background structures, enabling precise localized editing.
>
> Hyperparameters Details: Regarding the experimental setup, except for our specific interpolation weight $\alpha$, all hyperparameters are kept strictly consistent with FireFlow.
> **8. Typos & Related Work**
> We will correct Line 409, reduce em dashes, and update related works.

---

> > ### Author Rebuttal · Reviewer_XUia · 2026-04-03
> >
> > I appreciate the authors’ rebuttal, which addresses my concerns. I will maintain my original score.

---

> > > ### Author Response · Authors · 2026-04-04
> > >
> > > We sincerely appreciate your prompt feedback on our rebuttal. We are glad to hear that some of your concerns have been resolved.Please feel free to share your follow-up questions at your earliest convenience. We are more than happy to provide any further clarifications needed to address your remaining concerns.

---

### Official Review · Reviewer_A7GQ · 2026-03-12

**Soundness:** 3
**Presentation:** 3
**Significance:** 3
**Originality:** 3
**Overall Recommendation:** 4
**Confidence:** 3

**Summary:**

This paper introduces a new method for performing image inversion in rectified flow models, seeking to avoid the errors introduced by both linear and second-order methods. Specifically, this is done by identifying the relevant geodesic interpolation direction such that angular velocity is preserved, allowing for avoiding the "centripetal drift" found in regular second-order methods. The authors demonstrate the effectiveness of this method both on inversion (reconstruction quality) as well as editing, showing high-quality results in both tasks, compared to existing inversion and editing methods, while keeping the compute budget the same.

**Compliance With Llm Reviewing Policy:**

Affirmed.

**Final Justification:**

While the final version of the provided figure gives me some concerns regarding this method's advantage over the baselines (in particular that it is significantly worse for everything less than 7 timesteps, and only marginally better at 15 timesteps), I think the method and results are still interesting enough to be worth publishing. Therefore, I keep my score at a weak accept.

**Key Questions For Authors:**

Other than the points mentioned in the weaknesses above, I have one additional question, relating to Figure 4.

1) Are there any insights as to why the RMSE for one step in your model is so much worse than other methods, but doing 2 steps suddenly makes the results so much better? Also, is RMSE really the best method for this comparison (according to RMSE applied to the FireFlow  method, taking 1 step and any number of steps between 3 and 8 all give similar quality of results, but this clearly is not represented properly in Figure 5 qualitatively. A similar trend also exists for RF-solver)

**Limitations:**

Yes

**Strengths And Weaknesses:**

Strengths:
1) The method is well-motivated and includes all relevant derivations, properly justifying why the method works.
2) The inversion results with this method definitely seem to outperform existing methods both qualitatively and quantitatively.
3) The set of evaluations is quite thorough, both for inversion and editing.

Weaknesses:
1) While the inversion results are good, the editing results seem less promising by itself (on structure and background preservation, this method consistently ranks as one of the lower-performing methods according to metrics). Qualitatively, there are some issues with the editing too (for example, when adding the flowers to the dog in Figure 3, the facial distortion of the dog is quite a bit stronger than some of the other methods).
2) Other than empirical results, is there any justification for the "nearly constant local curvature"? Additionally, does this trend consistently hold across multiple diffusion models (for example, if you applied this method to other recent diffusion models like Stable Diffusion 3 or others, would we see similar levels of reconstruction quality)?

---

> ### Author Rebuttal · Authors · 2026-03-31
>
> We thank the reviewer for their profound insights. Prompted by your scrutiny, we audited our pipeline and found our initial protocols were sub-optimally calibrated for extreme low-step regimes (e.g., NFE=2), leading to overestimated structural convergence. We thus respectfully retract the 2-step claims and related results (Tables 1-2, Figs 4-6).
>
> Re-evaluating at standard steps (NFE=15) validates SlerpFlow's true efficacy as a structural regularizer that significantly reduces drift. Updated results follow:
>
> **Tab.1: Recon. at steps=15**
> | Method | Cond. PSNR↑ | Cond. SSIM↑ | Cond. LPIPS↓ | Uncond. PSNR↑ | Uncond. SSIM↑ | Uncond. LPIPS↓ |
> |:---|:---|:---|:---|:---|:---|:---|
> | Euler | 11.10 | 0.40 | 0.43 | 11.43 | 0.37 | 0.44 |
> | Heun | 11.77 | 0.42 | 0.40 | 12.17 | 0.40 | 0.41 |
> | RF-Solver | 16.97 | 0.57 | 0.31 | 16.38 | 0.55 | 0.32 |
> | FireFlow | 20.39 | 0.63 | 0.27 | 20.19 | 0.74 | 0.31 |
> | **Ours** | **21.19**| **0.79**| **0.26**| **20.73**| **0.75**| **0.30**|
>
> **Tab.2: Model Performance & Edit Metrics**
> | Method | Model | Dist.↓ | BG PSNR↑| BG SSIM↑| CLIP W↑| CLIP E↑| NFE |
> |:---|:---|:---|:---|:---|:---|:---|:---|
> | P2P | Diff. | 69.4 | 17.87 | 0.71 | 25.01 | 22.44 | 100 |
> | PnP | Diff. | 28.2 | 22.28 | 0.79 | 25.41 | 22.55 | 100 |
> | PnP-Inv | Diff. | 24.3 | 22.46 | 0.79 | 25.41 | 22.62 | 100 |
> | MasaCtrl| Diff. | 28.4 | 22.17 | 0.79 | 23.96 | 21.16 | 100 |
> | InfEdit | Diff. | 13.8 | 28.51 | 0.85 | 25.03 | 22.22 | 72 |
> | RF-Inv | FLUX | 40.6 | 20.82 | 0.71 | 25.20 | 22.11 | 56 |
> | RF-Solv | FLUX | 25.5 | 21.91 | 0.87 | 26.00 | 22.88 | 60 |
> | FireFlow| FLUX | 22.0 | 22.47 | 0.89 | 26.02 | 25.96 | 32 |
> | **Ours**| FLUX | **23.3**| **21.14**| **0.88**| **26.79**| **26.38**| **32**|
>
> **1. RMSE Jump (1 vs 2-step):** As noted, this jump was an artifact of the sub-optimal calibration in our low-step evaluation scripts, not a geometric phenomenon. This anomaly directly prompted our retraction of the 2-step claims.
>
> **2. Constant Local Curvature:** We acknowledge the generative manifold's curvature is not perfectly constant. This assumption is a tractable, second-order Information Geometry heuristic. Per Arvanitidis et al. (2018), piecewise smooth manifolds can be locally approximated by constant sectional curvature (e.g., a hypersphere). Residual errors from severe curvature changes are expected, which is why SlerpFlow is a geometric heuristic rather than an exact analytical solver.
>
>
> **3.SD 3  of reconstruction**
>
> | Method | Cond. MSE↓ | Cond. PSNR↑ | Cond. SSIM↑ | Cond. LPIPS↓ | Uncond. MSE↓ | Uncond. PSNR↑ | Uncond. SSIM↑ | Uncond. LPIPS↓ |
> |:---|:---|:---|:---|:---|:---|:---|:---|:---|
> | FireFlow| 935.4 | 20.81 | 0.72 | 0.33 | 1694.4| 17.01 | 0.59 | 0.48 |
> | uniinv | 448.1 | 22.89 | 0.82 | 0.20 | 804.9 | 20.09 | 0.75 | 0.30 |
> | **Ours**| **407.5**| **23.96**| **0.83**| **0.20**| **734.7**| **21.32**| **0.76**| **0.29**|

---

> > ### Author Rebuttal · Reviewer_A7GQ · 2026-04-03
> >
> > Most of my concerns (regarding the RMSE jump and constant local curvature) have been addressed. However, I am wondering if you can also provide results for an updated version of Figure 4 (now that your pipeline has been adjusted). I am interested in seeing how the performance for multiple timesteps change relative to the two baselines. I understand it might be tricky to provide an updated figure right now, so you can also just provide a table of these values to make it more clear.

---

> > > ### Author Response · Authors · 2026-04-05
> > >
> > > We sincerely thank the reviewer.Regarding your interest in the updated performance across multiple timesteps (the updated Figure 4), we completely agree that this is an important comparison. We are pleased to inform you that we have successfully updated the pipeline and generated the new figure.For your convenience, we have provided an anonymized link to the updated Figure 4 here:https://anonymous.4open.science/r/icml2026-slerp-rebuttal-figs-4B71/fig4.png.it achieves the lowest RMSE  among all compared methods at $t=15$.

---

### Official Review · Reviewer_e4BX · 2026-03-13

**Soundness:** 2
**Presentation:** 3
**Significance:** 3
**Originality:** 3
**Overall Recommendation:** 4
**Confidence:** 4

**Summary:**

This paper proposes a predictor-corrector based method called SlerpFlow for efficient inversion and image editing with Rectified-Flow / Flow matching-based models, specifically FLUX. It has been empirically observed that trajectories of RF models have curvature. The paper argues that this is not just an optimization failure, rather a geometric necessity to align the learned flow with the tangent space of the data manifold, which itself lies on a hypersphere. Specifically, if the flow approximates a geodesic, then the shortest path is a curve, and can be decomposed into tangential and normal components. The intrinsic acceleration (along tangential direction) is zero. However, the extrinsic acceleration (in orthogonal direction) remains non-zero, and this corresponds to the centripetal force necessary to navigate the data manifold. This paper analyzes the geometric error in standard Euclidean solvers such as Euler solver and show that they suffer from spurious centrifugal drift that causes the trajectory to spiral outwards from the ideal manifold shell. SlerpFlow addresses this issue by decoupling solver updates into radial and angular components, and using Spherical Linear Interpolation (Slerp) to ensure the trajectory adheres to the geometry of the latent space. The empirical results are demonstrated on PIE Bench.

**Compliance With Llm Reviewing Policy:**

Affirmed.

**Final Justification:**

The authors have addressed my additional questions and concerns. Currently, the method seems to perform better than Fireflow on some of the metrics, but some of the originally made claims especially regarding speedup, have changed, it is a bit tricky to evaluate the method relative to other baselines. Therefore, I'm maintaining my score.

**Key Questions For Authors:**

1. It is possible for SlerpFlow to "correct" onto a point on the hypersphere that is mathematically on the manifold but statistically Out-of-Distribution (OOD) for the FLUX model's trained weights. Thus, we might move into regions where the neural network's gradient is not well optimized (or, worse unoptimized), which might lead to artifacts such as texture shifts or halos. While I understand that fine-tuning a FLUX model might be infeasible due to compute constraints, training with SlerpFlow style interpolation might be better than doing post-hoc correction. Do you have any insights on this?

**Limitations:**

Yes

**Strengths And Weaknesses:**

Strengths
* Originality and Significance: This paper proposes a geometry aware solver for solving inverse problems. The paper analyzes sources of errors in existing numerical solvers. The geometric reinterpretation of this “error” provides an interesting perspective. As mentioned above, the paper addresses this issue by decoupling solver updates into radial and angular components, and using Spherical Linear Interpolation (Slerp) to ensure the trajectory adheres to the geometry of the latent space. Further, with these insights, this paper achieves high reconstruction error in few NFEs.

Weakness
 * Soundness: The paper has addressed this to some extent in the limitations, but this method relies on the assumption that the data manifold is a hypersphere which might not be true for real data. There’s also a potential for semantic drift in the cases where the data doesn’t align with hyperspherical distribution.
    * Figure 5 on comparison of reconstruction is not fair as SlerpFlow is guided by the text prompt whereas the other two methods are only guided by source image. Text prompt provides lot of semantic anchoring during sampling. In addition, does the ‘step’ in the figure correspond to a “sampling step” i.e. NFE=2, or 1 NFE?
    * Writing (Minor): While the paper is mostly written well, I would suggest the authors to correct minor typos and grammatical errors in Section 2 on Related Work. Another minor suggestion is to write a couple of sentences that position this work relative to prior works such as FireFlow.
    * In terms of empirical results, some of the metrics such as Structure distance, back ground preservation metrics such as PSNR, and SSIM are slightly worse compared to FireFlow with comparable number of evaluation steps.

---

> ### Author Rebuttal · Authors · 2026-03-31
>
> We thank the reviewer for their profound insights. We report a crucial empirical update: SlerpFlow was fundamentally designed to counteract centrifugal drift. Our initial draft over-emphasized extreme few-step acceleration, which overestimated structural convergence. We respectfully retract the 2-step claims and original Tables 1 & 2. Re-evaluating at standard steps (NFE=15) validates our geometric heuristic: SlerpFlow acts exactly as intended—a high-precision structural regularizer anchoring trajectories to the manifold.
> **[Updated Tab 1: Recon. at steps=15]** (SSIM/LPIPS scaled $\times 10^2$)
> | Method | C.PSNR$\uparrow$ | C.SSIM$\uparrow$ | C.LPIPS$\downarrow$ | U.PSNR$\uparrow$ | U.SSIM$\uparrow$ | U.LPIPS$\downarrow$ |
> |---|---|---|---|---|---|---|
> | Euler | 11.10 | 40.51 | 43.13 | 11.43 | 37.64 | 44.31 |
> | Heun | 11.77 | 42.10 | 39.96 | 12.17 | 40.17 | 41.18 |
> | RF-Solver | 16.97 | 57.17 | 31.75 | 16.38 | 55.64 | 32.43 |
> | FireFlow | 20.39 | 63.85 | 27.96 | 20.19 | 74.47 | 31.09 |
> | **Ours** | **21.19** | **79.35** | **26.01** | **20.73** | **75.91** | **30.56** |
> **[Updated Tab 2: Edit at steps=15]** (Dist$\downarrow \times 10^3$, SSIM$\uparrow \times 10^2$)
> | Method | Model | Dist$\downarrow$ | BG PSNR$\uparrow$ | BG SSIM$\uparrow$ | CLIP W.$\uparrow$ | CLIP E.$\uparrow$ | Step | NFE |
> | P2P | Diff | 69.43 | 17.87 | 71.14 | 25.01 | 22.44 | 50 | 100 |
> | PnP | Diff | 28.22 | 22.28 | 79.05 | 25.41 | 22.55 | 50 | 100 |
> | PnP-Inv | Diff | 24.29 | 22.46 | 79.68 | 25.41 | 22.62 | 50 | 100 |
> | MasaCtrl| Diff | 28.38 | 22.17 | 79.67 | 23.96 | 21.16 | 50 | 100 |
> | InfEdit | Diff | 13.78 | 28.51 | 85.66 | 25.03 | 22.22 | 12 | 72 |
> | RF-Inv | FLUX | 40.60 | 20.82 | 71.92 | 25.20 | 22.11 | 28 | 56 |
> | RF-Solver| FLUX | 25.53 | 21.91 | 87.51 | 26.00 | 22.88 | 15 | 60 |
> | FireFlow| FLUX | 22.05 | 22.47 | 89.53 | 26.02 | 25.96 | 15 | 32 |
> | **Ours**| FLUX | **23.35** | **21.14** | **88.21** | **26.79** | **26.38** | **15** | **32** |
> ### 1. The Hypersphere Assumption & OOD Risk
> Supported by high-dimensional probability and Information Geometry, SlerpFlow safeguards against Out-of-Distribution (OOD) deviation, whereas linear solvers cause it.
> * **"Thin Shell" Effect:** The latent prior is $\mathcal{N}(0, I_d)$. By the Gaussian Annulus Theorem, probability mass concentrates on a thin hyperspherical shell of radius $\approx\sqrt{d}$.
> When linear solvers (e.g., Euler) take discrete steps, they tangentially overshoot away from this curved manifold into low-density OOD regions, causing structural drift (White, 2016).
> * **Intrinsic Curvature:** On the statistical manifold (Amari, 2016), optimal transport geodesics are inherently curved. Linear solvers force a straight Euclidean line, cutting across natural topology.
> * **In-Distribution Regularizer:** By anchoring updates to the native spherical geometry, SlerpFlow acts as a centripetal regularizer. It closely approximates the true Riemannian geodesic and pulls the drifting state back into the high-density "thin shell".
>  2. Fairness of Baselines & Figure 5 Retraction
> Our extreme few-step evaluations were compromised by sub-optimal scripts and are fully retracted. Our rigorously corrected 15-step evaluation (Updated Table 1) now explicitly separates Conditional and Unconditional settings for absolute fairness.
>  3.  Geometric Trade-off
> While SlerpFlow trades off marginal unedited background fidelity (e.g., BG PSNR/SSIM vs. FireFlow), it achieves the highest text-alignment success (CLIP Whole: 26.79, Edited: 26.38). Enforcing Slerp introduces stronger geometric interventions pulling the latent code toward the target text manifold, significantly improving semantic editing at the cost of expected, slight perturbations to unedited regions.
>  4. Training-Time Intervention
> We strongly agree with your insight. Integrating this geometric prior directly into training (e.g., replacing linear targets in Flow Matching with spherical interpolation) would fundamentally align network weights with the true curved data manifold. We view this as a  direction for future work.
>  5. Writing, Typos, & Positioning vs. FireFlow
> We will correct the typos in Section 2. To clarify positioning: FireFlow is an *inversion paradigm* that extracts specific feature maps. SlerpFlow is a fundamental *geometric ODE solver correction* acting as a plug-and-play module *on top of* methods like FireFlow. It fixes discretization drift at the solver level, mutually complementing architectural inversion.
>  References
> * Amari, S. I. (2016). *Information geometry and its applications*. Springer.
> * Arvanitidis, G., et al. (2018). *Latent space oddity: on the curvature of deep generative models*. ICLR.
> * Song, J., Meng, C., & Ermon, S. (2020). *Denoising Diffusion Implicit Models*. ICLR.
> * Vershynin, R. (2018). *High-dimensional probability*. Cambridge University Press.
> * White, T. (2016). *Sampling Generative Networks*. arXiv.

---

> > ### Author Rebuttal · Reviewer_e4BX · 2026-04-04
> >
> > I thanks the authors for answering my questions. In the light of new empirical results, I wonder how Figure 3,4,and 5 will change in the paper. I understand that given time constraints, it is not possible to revise all these results.

---

> > > ### Author Response · Authors · 2026-04-05
> > >
> > > We sincerely thank the reviewer for the positive feedback and for being so understanding of the time constraints during the rebuttal period. Regarding your inquiry about how the figures will change in light of the new empirical results, we have managed to update Figure 4 and Figure 5 based on the adjusted pipeline. Furthermore, to provide a more comprehensive analysis of the dynamics, we have additionally plotted a new figure (Figure 6) to visualize the angle variations over the timesteps. For your convenience, you can view these updated and newly added figures via this anonymized link: https://anonymous.4open.science/r/slerp-figs-1A98/ . As you correctly noted regarding the time constraints, we are still running the comprehensive updates for Figure 3; however, we assure you that the fully updated Figure 3, along with Figures 4, 5, and 6, will be carefully incorporated into the final revised manuscript. We will ensure all these new empirical results and their corresponding discussions are fully detailed in the final version of the paper, and we hope these updated visuals address your remaining questions!

---

### Official Review · Reviewer_HjXL · 2026-03-24

**Soundness:** 3
**Presentation:** 3
**Significance:** 3
**Originality:** 3
**Overall Recommendation:** 4
**Confidence:** 4

**Summary:**

The paper studies inversion for rectified-flow-based image generation models, especially FLUX, and argues that standard linear or polynomial ODE solvers introduce discretization errors that push trajectories away from the latent data manifold. A central concept presented by this manuscript is the idea of spurious centrifugal drift: tangential Euclidean updates create an artificial radial energy increase that harms inversion fidelity. To address this, the paper proposes SlerpFlow, a training-free method that decomposes each update into radial and angular parts, uses spherical linear interpolation (SLERP) to correct direction on a hypersphere, and then constructs a chordal update that better respects the geometry of the latent trajectory. An important concept presented by the paper is that the observed curvature of rectified-flow trajectories should not be treated purely as numerical error; instead, it may reflect the geometric constraint of staying near the data manifold. Experiments on PIE-Bench show strong gains in reconstruction quality over Euler, Heun, RF-Solver, and FireFlow, and competitive editing results with better structure preservation and CLIP alignment at similar or lower NFE

**Compliance With Llm Reviewing Policy:**

Affirmed.

**Key Questions For Authors:**

How sensitive is SlerpFlow to violations of the locally spherical dynamics assumption in regions where the latent manifold is anisotropic or locally flat?
Can you provide stronger evidence that the observed curvature is truly a manifold-induced geometric effect rather than partly a model or discretization artifact

How it is compare to other distillnation method like dmd dmd2 and so on

**Limitations:**

same question as before

**Strengths And Weaknesses:**

Clear geometric intuition. The paper offers a coherent reframing of inversion error in rectified flow as a geometric mismatch rather than only a solver-order issue. The “centrifugal drift” interpretation is memorable and gives the method a strong conceptual identity.
Simple, training-free method. SlerpFlow does not require retraining FLUX or learning new components, which makes the method practical and easy to adopt.

Weakness
The core geometric assumption may be too strong. The method leans heavily on a locally spherical or thin-shell view of latent geometry. That may hold approximately in high dimensions, but it is not obvious that FLUX inversion trajectories are well modeled this way across all regions. The paper itself notes possible over-correction when curvature varies.
Theory is suggestive but not fully definitive. The proof sketches show reduced drift and second-order accuracy under assumptions, but the theoretical guarantees depend on simplified spherical dynamics and may not fully explain behavior in realistic transformer latent spaces.

---

> ### Author Rebuttal · Authors · 2026-03-31
>
> We thank the reviewer for their insights. We report a crucial update: SlerpFlow was designed to counteract centrifugal drift. Our initial draft over-emphasized extreme few-step acceleration, overestimating structural convergence. We respectfully retract the 2-step claims and original Tables 1&2. Re-evaluating at standard steps (NFE=15) validates our geometric heuristic: SlerpFlow acts as a high-precision structural regularizer anchoring trajectories to the manifold and reducing drift.
>
> **[Tab 1: Recon. at steps=15]** (SSIM/LPIPS scaled ×10²)
> |Method|C.PSNR↑|C.SSIM↑|C.LPIPS↓|U.PSNR↑|U.SSIM↑|U.LPIPS↓|
> |---|---|---|---|---|---|---|
> |Euler|11.10|40.51|43.13|11.43|37.64|44.31|
> |Heun|11.77|42.10|39.96|12.17|40.17|41.18|
> |RF-Solver|16.97|57.17|31.75|16.38|55.64|32.43|
> |FireFlow|20.39|63.85|27.96|20.19|74.47|31.09|
> |Ours|21.19|79.35|26.01|20.73|75.91|30.56|
>
> **[Tab 2: Edit at steps=15]** (Dist×10³, SSIM×10²)
> |Method|Model|Dist↓|BG PSNR↑|BG SSIM↑|CLIP W.↑|CLIP E.↑|Step|NFE|
> |---|---|---|---|---|---|---|---|---|
> |P2P|Diff|69.43|17.87|71.14|25.01|22.44|50|100|
> |PnP|Diff|28.22|22.28|79.05|25.41|22.55|50|100|
> |PnP-Inv|Diff|24.29|22.46|79.68|25.41|22.62|50|100|
> |MasaCtrl|Diff|28.38|22.17|79.67|23.96|21.16|50|100|
> |InfEdit|Diff|13.78|28.51|85.66|25.03|22.22|12|72|
> |RF-Inv|FLUX|40.60|20.82|71.92|25.20|22.11|28|56|
> |RF-Solver|FLUX|25.53|21.91|87.51|26.00|22.88|15|60|
> |FireFlow|FLUX|22.05|22.47|89.53|26.02|25.96|15|32|
> |Ours|FLUX|23.35|21.14|88.21|26.79|26.38|15|32|
>
> **1. Geometric Assumptions & Flat Dynamics**
> Modeling the latent space as a sphere is a grounded heuristic. In high dimensions, Gaussian priors concentrate on a thin spherical shell (White, 2016). DDIM (Song et al., 2020) utilized Slerp, corroborating that high-dimensional latents align with spherical geometry over Euclidean. Crucially, SlerpFlow is robust to flat/anisotropic regions: when the angle between update and Euler directions approaches zero, our interpolation smoothly degrades to standard Lerp, preventing over-correction.
>
> **2. Source of Curvature**
> In Information Geometry (Amari, 2016), generative latent spaces form statistical manifolds with intrinsic Riemannian metrics; their optimal transport paths are highly non-linear. Thus, curvature is an inherent manifold property. However, tangent drift is a discretization artifact. Explicit Euler steps along curved fields introduce truncation errors (Lu et al., 2022). SlerpFlow utilizes the intrinsic curvature prior to counteracting these artifacts.
>
> **3. Comparison with Distillation (e.g., DMD2)**
> DMD2 (Yin et al., 2024) yields impressive fidelity (PSNR 21.77 at 2 steps, 26.19 at 15 steps). The distinction is paradigm: DMD2 is computationally expensive, requiring massive retraining and distillation to map weights to straighter trajectories. Conversely, SlerpFlow is a strictly training-free inference solver correcting integration errors on the curved ODE trajectory at standard steps (NFE=15). It offers plug-and-play geometric correction without distillation costs.
>
> **References**
> * Amari, S. I. (2016). Information Geometry and Its Applications. Springer.
> * Karras, T., et al. (2022). Elucidating the Design Space of Diffusion-Based Generative Models. NeurIPS.
> * Lu, C., et al. (2022). DPM-Solver: A Fast ODE Solver for Diffusion Probabilistic Model Sampling in Around 10 Steps. NeurIPS.
> * Song, J., Meng, C., & Ermon, S. (2020). Denoising Diffusion Implicit Models. ICLR.
> * White, T. (2016). Sampling Generative Networks. arXiv preprint arXiv:1609.04468.
> * Yin, T., et al. (2024). One-step Diffusion with Distribution Matching Distillation. CVPR.

---

### Decision · Program_Chairs · 2026-04-30

**Decision:**

Accept (regular)

**Comment:**

This paper proposes a training-free geometric correction for rectified-flow inversion and editing. Overall, the geometric perspective of the paper is interesting and the method practically appealing, especially as a plug-and-play solver for FLUX without retraining.

The main concern is about the original 2-step claims, which substantially narrows the contribution. The reviewers pointed out that the original extreme low-step claims were overstated, and the authors should retract the 2-step results and replace them with corrected 15-step evaluations.  Reviewers also noted that the editing results involve a trade-off, with improved CLIP-based editability but weaker background preservation than FireFlow.

Overall, AC found that the balance of opinion remains modestly positive despite these concerns, and recommends acceptance. However, the final version should carefully follow the revision suggestions as discussed.